# Activity-based protein profiling as a robust method for enzyme identification and screening in extremophilic Archaea

Susanne Zweerink[1,*], Verena Kallnik[2,*], Sabrina Ninck[1], Sabrina Nickel[1], Julia Verheyen[2], Marcel Blum[2], Alexander Wagner[3], Ingo Feldmann[4], Albert Sickmann[4], Sonja-Verena Albers[3], Christopher Bräsen[2], Farnusch Kaschani[1], Bettina Siebers[2] & Markus Kaiser[1]

Archaea are characterized by a unique life style in often environmental extremes but their thorough investigation is currently hampered by a limited set of suitable *in vivo* research methodologies. Here, we demonstrate that *in vivo* activity-based protein profiling (ABPP) may be used to sensitively detect either native or heterogeneously expressed active enzymes in living archaea even under these extreme conditions. In combination with the development of a genetically engineered archaeal screening strain, ABPP can furthermore be used in functional enzyme screenings from (meta)genome samples. We anticipate that our ABPP approach may therefore find application in basic archaeal research but also in the discovery of novel enzymes from (meta)genome libraries.

[1] Chemische Biologie, ZMB, Fakultät für Biologie, Universität Duisburg-Essen, Universitätsstr. 2, 45117 Essen, Germany. [2] Molekulare Enzymtechnologie und Biochemie, Biofilm Centre, ZWU, Fakultät für Chemie, Universität Duisburg-Essen, Universitätsstr. 2, 45117 Essen, Germany. [3] Molecular Biology of Archaea, Institute of Biology II—Microbiology, University of Freiburg, Schänzlestr. 1, 79104 Freiburg, Germany. [4] Leibniz-Institut für Analytische Wissenschaften—ISAS e.V., Bunsen-Kirchhoff-Str. 11, 44139 Dortmund, Germany. * These authors contributed equally to this work. Correspondence and requests for materials should be addressed to B.S. (email: bettina.siebers@uni-due.de) or to M.K. (email: markus.kaiser@uni-due.de).

Enzymes play decisive roles in modern biotechnology[1,2] and many different approaches such as enzyme engineering, *de novo* design or directed enzyme evolution for customized enzyme generation have been developed[2–6]. Despite these advances, systematic functional screenings of organisms have nevertheless remained an important approach for biocatalyst discovery, in particular in combination with next generation sequencing environmental metagenomic approaches that enable sequence-based or together with appropriate screening strategies functional identification of novel biocatalysts[7,8].

Archaea form one of the three domains of life and have recently be proposed as the progenitors of eukaryotic cells (Fig. 1a)[9–11]. Many of the so far cultured archaea thrive under environmental extremes of temperature, pH or salinity. They combine biochemical and cellular properties of bacteria and eukarya but also display exceptional features like the use of distinct membrane lipids or an exclusive cell wall composition[9,12]. Although their overall metabolic complexity resembles bacteria and lower eukarya, they lack many of the classical central metabolic pathways but instead harbour modified variants that involve enzymes with only weak or even no similarity to bacterial or eukaryotic counterparts[13]. These unique metabolic properties and their resistance to extreme conditions turn enzymes from these species into an exclusive but until now largely unexplored branch of biocatalytic biodiversity. Moreover, many of the identified open reading frames in archaeal genomes are of so far unassigned function indicating further uncharted biocatalytic diversity[14,15].

However, only a few archaeal species are easily cultivable and their extremophilic growth requirements often impede the use of standard biochemical methodologies. Therefore, enzymes from archaea are typically studied after expression in a heterologous host, mostly *E. coli*. For certain archaeal proteins, this is however not possible due to low expression rates, inaccurate protein folding or the lack of functionally important post-translational protein modifications in these hosts. In addition, the identification of archaeal enzymes directly from genome samples in bacterial host-based functional screening assays is challenging as archaeal promotor structures differ from bacterial ones but show similarities to those found in eukarya. A direct *in vivo* investigation of archaeal enzymes in an archaeal strain or an archaeal host conferring (over)expression of the respective proteins would therefore represent a valuable alternative but requires (i) a technically simple and sensitive technology to detect active enzymes and (ii) a methodology to heterologously express proteins in an archaeal host. In the present study, we propose that activity-based protein profiling (ABPP) is such a sensitive detection technology while the archaeal species *Sulfolobus acidocaldarius* represents such a suitable expression host.

In bacteria and eukaryotes, ABPP is an established methodology for measuring the activity state of enzymes under physiological conditions[16–19]. It relies on the use of activity-based probes (ABPs) that irreversibly label active enzymes, either *in vitro* or even *in vivo*. The ABPs are built up from a 'warhead' which is, a bioreactive enzyme inhibitor residue that reacts with the active site of an enzyme in an activity-dependent manner, a linker moiety and finally a reporter tag for visualization and/or detection of labelled proteins. In *in vivo* ABPP, the reporter tag of ABPs is often an alkyne or azide residue because these relatively small residues have only weak detrimental effects on cell permeability[20]. The use of such probes in ABPP experiments requires a two-step labelling methodology: in the first step, the ABP is added *in vivo* to the biological system to enable ABPP labelling; in the second step, a 'standard' reporter tag such as a fluorophore moiety is attached after cell lysis *via* bioorthogonal conjugation chemistries such as a Staudinger ligation or a Huisgen-type Cu(I)-catalysed (3 + 2) cycloaddition (also known as 'Click' reaction) to enable in-gel detection of labelled proteins[21,22]. Besides fluorophores, some 'clickable' reporter tags also harbour an additional biotin residue for enrichment of labelled proteins via an avidin affinity purification and subsequent in gel-detection.

Many ABPs display enzyme family/class-specific labelling properties and use of these probes will detect only representatives of these enzyme classes/families in living organisms. For example, serine hydrolase, cysteine or threonine protease, tyrosine phosphatase, glycosidase or ATPase and kinase-specific ABPs are available[16–19]. Consequently, *in vivo* application of such ABPs to archaea could be used to detect and functionally annotate enzymes of a selected family/class in a target-oriented manner (Fig. 1b)[23]. The challenge for such an approach is, however, to establish suitable conditions and ABPs that enable ABPP under the harsh extremophilic growth conditions. To the best of our knowledge, no *in vivo* labelling of archaea has yet been performed; only an ABPP of an archaeal lysate has already been performed that however did not deliver robust target identifications[24].

The aerobic, thermoacidophilic crenarchaeon *S. acidocaldarius* was isolated as one of the first (hyper)thermophiles from acidic hot springs at the Yellowstone National Park and grows optimally at 75–80 °C and a pH of 2–3 (ref. 25). The genome sequence as well as a comprehensive genetic toolbox is available for this archaeal model organism[26,27]. Its robust and versatile genetic system enables the construction of in frame markerless deletion mutants, ectopic integration of foreign DNA and an effective homologous and also heterologous expression, since the internal pH of the organism was reported to be around 6.5. Thus, *S. acidocaldarius* is a suitable system to study homologous and heterologously expressed proteins from (hyper)thermophilic organisms.

In this present work we will demonstrate that *in vivo* ABPP represents a technically simple, sensitive and 'extremophile-compatible' technology to detect even low levels of active enzymes in Archaea. To this end, we provide a proof-of-concept study with archaeal serine hydrolases because they are (i) ubiquitously expressed in all domains of life (incl. archaea) and represent a well-characterized enzyme family which features an active serine residue critically involved in substrate hydrolysis[28,29], (ii) known to be robust and reliable targets in ABPP approaches[30,31] and (iii) biocatalysts of great biotechnological interest[32]. We furthermore show that ABPP enables the identification of serine hydrolases in thermoacidophilic and halophilic Archaea either in the native archaeal system or in an archaeal expression host.

## Results

**In vivo ABPP of Sulfolobus acidocaldarius.** The thermo-acidophile *S. acidocaldarius* grows optimally at a temperature of 78 °C and a pH of 2.0–3.0. To establish a suitable protocol for *in vivo* ABPP under these harsh conditions, we focused our attention on phosphonate-based serine hydrolase probes because these probes are among the best characterized ABPs and are known to display robust labelling[30,31]. Consequently, we synthesized a small set of phosphonate-based probes that all featured a terminal alkyne tag (schematically depicted as a '≡' in this study) for performing two-step click-ABPP experiments (for chemical structures of all probes, see Supplementary Fig. 1; their synthesis is reported in the chemical synthesis section of Supplementary Information). The resulting probe collection consisted of (i) the 'classical' Cravatt fluorophosphonate-based probe **FP≡** (**1**) (Fig. 2a)[31], (ii) a nitrophenol ethyl phosphonate derivative **NP≡** (**2**, Fig. 2a) representing a **FP≡** analogue with reduced reactivity[33], (iii) two nitrophenol phosphonate probes with hydrophobic residues on the warhead; this was either a phthalylalkyl moiety (**Pt-NP≡**, **3**) or a heptanyl chain

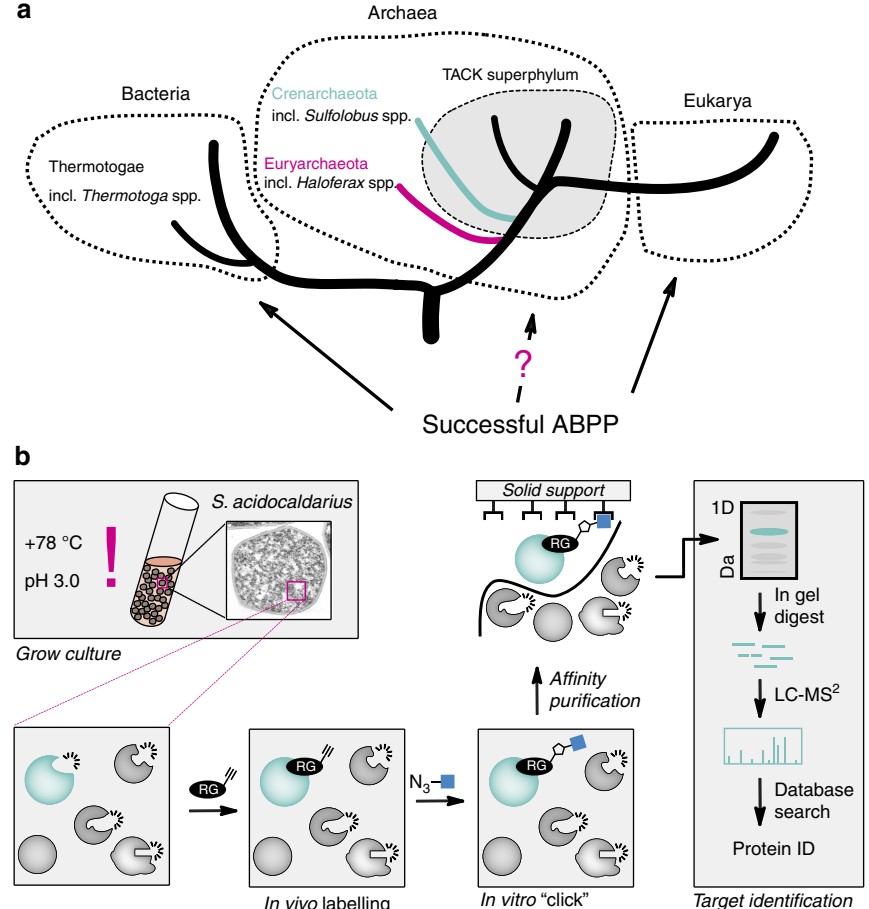

**Figure 1 | Overview of the three domains of life and the established ABPP work flow for *S. acidocaldarius*.** (**a**) The three domains Archaea, Bacteria and Eukarya as a tree of life. The bacterial and archaeal strains used in this communication are indicated in italic. (**b**) Envisaged two-step *in vivo* activity-based protein profiling work flow. Cells are grown in liquid culture and labelled *in vivo* with a click chemistry compatible probe. After *in vivo* labelling proteins are extracted and a suitable reporter or affinity purification tag is 'clicked' on to labelled proteins. Labelled proteins are then affinity purified and subsequently separated by SDS–PAGE. After excision of the labelled proteins from the gel and tryptic in-gel digestion the respective proteins are identified by LC–MS/MS. RG, reactive group.

(**Hp-NP≡**, **4**), (iv) a derivative that should serve as a negative labelling control, that is a **NP≡** derivative **NC–NP≡** (**5**) with a free acid functionality (no leaving group) on the phosphonate residue. All nitrophenol phosphonate probes were generated with a polyethylene glycol-based linker.

For gaining first insights into the labelling properties of these probes, we prepared *S. acidocaldarius* MW001 cell lysates (the mutant MW001 is a Δ*pyrEF* strain and is in this study referred to as the wildtype (WT) *S. acidocaldarius* strain) and performed *in vitro* labelling experiments at either 25 °C or 78 °C and at a pH of 8.0 which is the optimal pH for serine hydrolase labelling of lysates with **FP≡**. We also evaluated the effect of pre-incubation with or without 50 μM paraoxon (Supplementary Fig. 2). The two different temperatures were chosen to evaluate the impact of temperature on chemical probe stability while paraoxon, as a broad range esterase inhibitor, was included to identify paraoxon-sensitive esterases. Overall, we found comparable labelling patterns at 25 and 78 °C, even though some bands were slightly stronger and additional, very faintly labelled bands appeared at higher temperature indicating that higher temperatures induce an only weak increase in background labelling. The 'negative' control **NC–NP≡** displayed only weak background labelling at both temperatures. All other probes labelled at least one prominent band in the range of 38 kDa that

was sensitive to paraoxon pre-incubation. The most selective nitrophenol phosphonate-based probe was **NP≡** displaying only weak labelling of other proteins. Among the labelled bands was also a paraoxon-insensitive band at ca. 135 kDa that was also strongly labelled with the nitrophenol phosphonate probes **Pt-NP≡** and **Hp-NP≡**. The **FP≡** probe also labelled proteins in the 135 kDa region in addition to several other weakly labelled bands, confirming the reported broad range serine hydrolase labelling properties of **FP≡**. These results showed that the ABPP approach works *in vitro* under (hyper)thermophilic conditions in the physiological pH range. We decided to continue our studies with the apparently most selective probe, **NP≡**, as well as with **FP≡** as the probe labelling most proteins.

To establish the envisaged two-step *in vivo* ABPP labelling approach under thermoacidophilic conditions in which *Sulfolobus* species thrive (Fig. 1b), we incubated *S. acidocaldarius* MW001 cultures without or after 10 min pretreatment with either 50 μM paraoxon or 50 μM of PMSF (phenylmethylsulfonyl fluoride, an alternative broad range serine hydrolase inhibitor) at 78 °C and pH 3.0 for 2 h with 2 μM **NP≡** or **FP≡**. To differentiate between *in vivo* and *ex vivo* labelling, we used an established approach, that is, we lysed cells after probe treatment either in presence or absence of 2% SDS[34]. In the presence of 2% SDS proteins and enzymes should be denatured and inactive; therefore, labelling must have

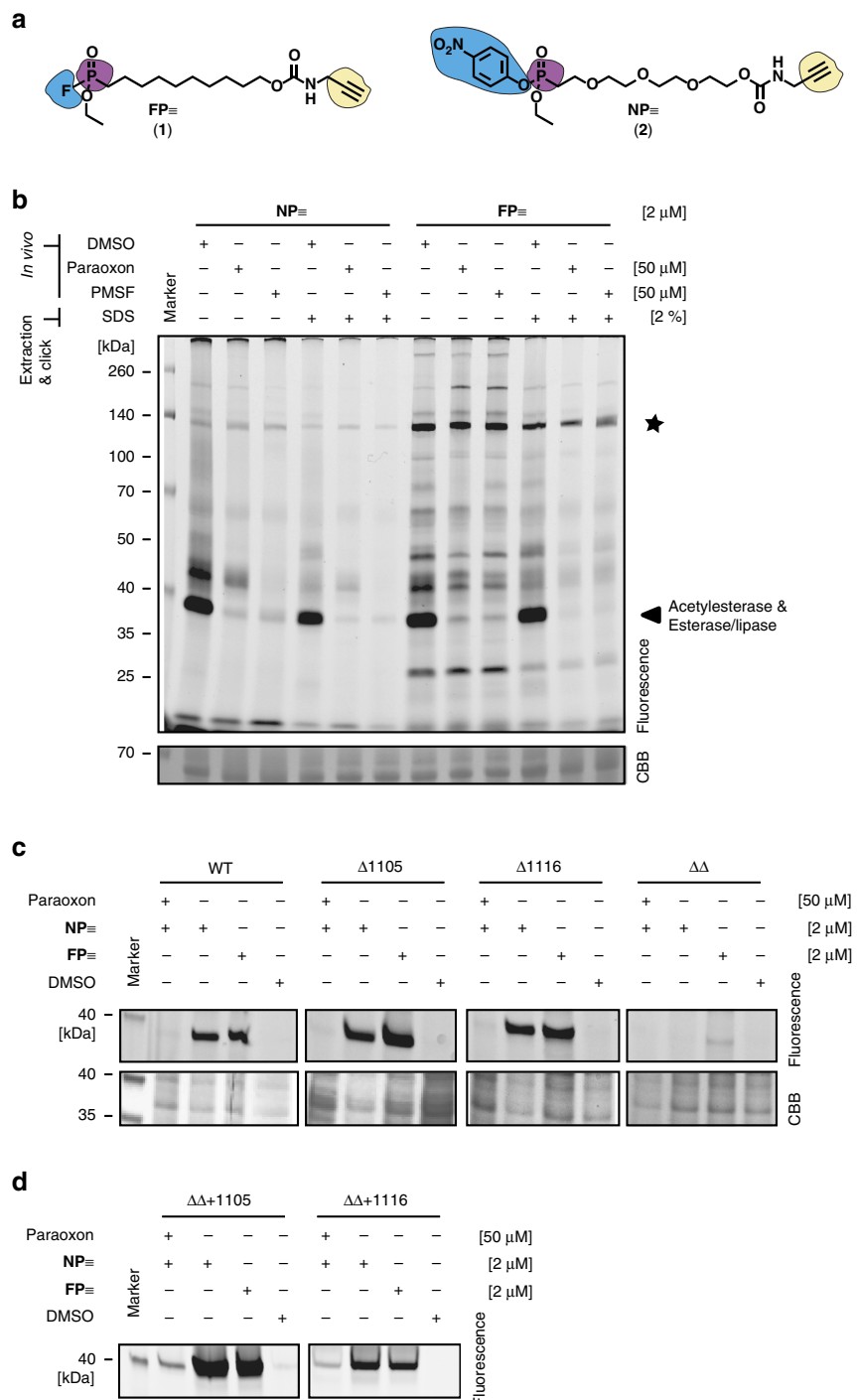

**Figure 2 | In vivo ABPP of Sulfolobus acidocaldarius.** (**a**) Chemical structure of the ABPs **FP**≡ and **NP**≡. The blue background depicts the reactive/leaving group, the purple colour marks the electrophilic centre and the yellow colour highlights the alkyne 'click' tag. (**b**) 1D SDS–PAGE of the in vivo profiled *S. acidocaldarius* MW001 proteome. Labelling with the two probes results in a similar labelling band pattern. **FP**≡ labelling is slightly less specific than **NP**≡. The strongest labelling is observed in the 35 kDa (arrowhead) and 140 kDa region (star) of the gel. The labelling was perfomed in vivo as extraction of proteins in the presence of 2% (w/v) SDS did not abolish the band pattern (the identity of targets in the 35 kDa region is based on Supplementary Fig. 3). (**c**) Analysis of *S. acidocaldarius* deletion strains for Saci_1105 (Δ1105) and Saci_1116 (Δ1116) show that labelling in the 35 kDa region is caused by these two proteins. In the double knockout strain (ΔΔ), the corresponding band is no more detected. WT corresponds to *S. acidocaldarius* MW001. (**d**) Complementation of the double deletion strain (ΔΔ) by overexpression of Saci_1105 and Saci_1116 (ΔΔ + 1105 or ΔΔ + 1116) restores the initial band pattern.

occurred in vivo. The lysates were then subjected to click chemistry labelling with **Rh-N₃**, in-gel separation and in-gel fluorescence visualization (for the chemical structure of **Rh-N₃**, see Supplementary Fig. 1). The gel analysis revealed that SDS treated samples

had slightly less background labelling but the overall labelling pattern remained constant and closely resembled the in vitro results (Fig. 2b). As observed in the in vitro labelling (Supplementary Fig. 2), the most prominent band was found at ca. 38 kDa and was

paraoxon-sensitive (arrowhead, Fig. 2b). A second less paraoxon-sensitive band was labelled at ca. 140 kDa (star, Fig. 2b). The 140 kDa band was also more pronounced if **FP≡** was used instead of **NP≡**. Besides, also several faintly labelled bands were apparent. PMSF pre-treatment resulted in similar labelling patterns as paraoxon pre-treatment. Therefore, in later experiments, only paraoxon was used as a labelling competitor. Overall, the employed procedure resulted in robust and reproducible *in vivo* labelling.

To reveal the identity of the labelled proteins, a gel-based target identification approach was performed (Supplementary Fig. 3). *S. acidocaldarius* MW001 was grown at 78 °C and pH 3.0 and labelled for 2 h with 2 μM **NP≡** or **FP≡**. The cells were lysed and a click reaction with **Rh-Biot-N₃** was performed on the extracted proteins, followed by an avidin affinity purification and SDS–polyacrylamide gel electrophoresis (SDS–PAGE) analysis (for the chemical structure of **Rh-Biot-N₃**, see Supplementary Fig. 1). Labelled proteins were excised from the gel, digested with trypsin and the obtained peptide mix analysed by LC–MS/MS. The raw mass spectra were then searched against a dedicated *S. acidocaldarius* database using the Andromeda search engine as implemented in MaxQuant[35]. As common for such experiments, we identified a large number of proteins which were co-affinity purified with the 'real' targets (Supplementary Data 1). After filtering the MaxQuant results for serine hydrolases, 10 distinct targets remained (Supplementary Fig. 3). Unexpectedly, the most prominent single band at 38 kDa turned out to be caused by simultaneous labelling of two esterases (reactive domain: Abhydrolase_3), which are acetylesterase Saci_1116 (MW 33.5 kDa, subsequently referred to as 1116) and esterase/lipase Saci_1105 (MW 34.6 kDa, subsequently referred to as 1105)[36,37]. In addition, the 140 kDa band was identified as a tricorn protease homologue (MW 116.6 kDa; reactive domain S41).

To test if the single band at 38 kDa indeed corresponded to the two esterases, a set of *S. acidocaldarius* deletion mutants was generated in which either one of the two (esterase/lipase Δ1105; acetyl esterase Δ1116) or both esterases (ΔΔ1105,1116) were deleted. **NP≡** and **FP≡** *in vivo* labelling of wildtype and both single mutant strains resulted in a band at 38 kDa, but no labelling was detected in the ΔΔ double mutant strain (Fig. 2c and Supplementary Fig. 4). This finding clearly indicates that labelling in the 38 kDa region is indeed caused by simultaneous labelling of both esterases. Comparable results were obtained for a similar *in vitro* labelling experiment (Supplementary Fig. 5).

Additional proof for the identity of the labelled proteins in the 38 kDa gel region was obtained by complementation studies, that is by overexpression of the lipase/esterase 1105 or the acetylesterase 1116 in the ΔΔ mutant strain. To this end, both genes were cloned into the expression vectors pSVAmZ-SH10 and pSVA1551, respectively, under the control of the inducible maltose (mal) promotor from *S acidocaldarius*[38]. After growth of the transformants in liquid culture in the presence of dextrin as an inducer, *in vivo* labelling at 78 °C and pH 3.0 with either **FP≡** or **NP≡** with or without preincubation with 50 μM paraoxon led to the re-appearance of the 38 kDa band (Fig. 2d). The successful expression was confirmed by MS analysis of the respective protein bands (Supplementary Data 1).

***In vivo*** **ABPP of further archaeal strains**. So far, we could show that ABPP with **FP≡** or **NP≡** is a suitable tool to display the activity of serine hydrolases in *S. acidocaldarius*. To demonstrate that the established *in vivo* conditions are sufficiently robust to enable a general ABPP workflow, we performed ABPP experiments with *S. solfataricus* (P2). *S. solfataricus* is a close relative of *S. acidocaldarius*, with similar growth requirements (78 °C, pH of 2.0–3.0). However, *S. solfataricus* has a significantly

increased metabolic versatility, which is well documented by an increased genome size (*S. acidocaldarius* (DSM639) 2,225,959 nucleotides[26], *S. solfataricus* (P2) 2,992,245 nucleotides[27,39]). A PFAM analysis predicted that this organism harbours overall 18 predicted serine hydrolases, many of them with structural homologies to *S. acidocaldarius* (Supplementary Data 2)[40].

ABPP of *S. solfataricus* was achieved by using the same experimental workflow as for *S. acidocaldarius* and in-gel fluorescence detection revealed labelling of many different proteins (Fig. 3a). In the most prominent band at ca. 35 kDa we identified several serine hydrolases (for example, the lipases LipP-1 and LipP-2 and the esterase Est). The protein Tri (SSO2098) at ca. 140 kDa was, as identified in *S. acidocaldarius*, a Tricorn protease homologue.

*S. solfataricus* LipP-1 (SSO2493) and *S. acidocaldarius* acetylesterase 1116 are close homologues (sequence identity 77%) while LipP-2 (SSO2521) only shows very moderate sequence homology to *S. acidocaldarius* 1116 (sequence identity 45%). The closest homologue of *S. acidocaldarius* lipase/esterase 1105 was *S. solfataricus* Est (SSO2517, sequence identity 72%)[41]. A pairwise alignment of the Tricorn protease sequences revealed an overall sequence identity of 65%. Besides these four proteins, we also identified two putative acylaminoacyl-peptidases (ApeH-1, SSO1419, MW 61.8 kDa and ApeH-2, SSO2141, MW 64.3 kDa), a serine aminopeptidase (SSO2518, MW 40.5 kDa), a proline iminopeptidase (Pip, SSO3115, MW 36.0 kDa), and 2 amidases (SSO2122, MW 55.7 kDa and GatA-1, SSO0765, MW 44.1 kDa). In total, we robustly identified 10 serine hydrolases out of the 18 predicted hydrolases (56%). This number is comparable to the identifications from *S. acidocaldarius* where we identified 10 serine hydrolases out of 17 predicted ones.

After we established the workflow under thermoacidophilic growth conditions, we were eager to analyse another extreme of life—high salt—in order to demonstrate the robustness of the new ABPP approach. Therefore, we profiled the evolutionary much more distant Archaeon *Haloferax volcanii* which belongs to the phylum Euryarchaeota (family halobacteriaceae). It requires high salt conditions (1.5–2.5 M salt) and a growth temperature of 30–55 °C. Genome analysis predicts a maximum of 22 serine hydrolases in this species (Supplementary Data 2)[41].

The corresponding ABPP experiment that was performed without any additional species-specific optimization of the established labelling methodology revealed a different labelling pattern compared to *S. acidocaldarius* or *S. solfataricus*, as expected for a different organism (Fig. 3b). LC–MS analysis enabled the identification of the ClpP-like protease sppA1 (HVO_0881, MW 35.2 kDa), the amidotransferase gatA (HVO_1054, MW 43.9 kDa) and the so far largely uncharacterized serine hydrolases HVO_2702 (MW 28.0 kDa), HVO_1591 (28.8 kDa) and HVO_1588 (MW 11.6 kDa). Altogether, these findings indicate that our established two-step ABPP protocol is robust for profiling and identification of active serine hydrolases in many different archaeal strains under extremes of temperature/pH and salinity.

***In vivo*** **ABPP of heterologously expressed serine hydrolases**. We then investigated if this approach can also be applied to detect heterologously expressed proteins in *S. acidocaldarius*. We used the ΔΔ *S. acidocaldarius* mutant as a heterologous archaeal screening host because the lack of the two prominent targets in the 38 kDa range combined with very little other labelling from **FP≡** or **NP≡** facilitates the gel-based identification of heterologously expressed serine hydrolases (especially in this mass range).

Two genes encoding the thermophilic serine hydrolases LipS (family Hydrolase 4; S33 serine aminopeptidase) and LipT

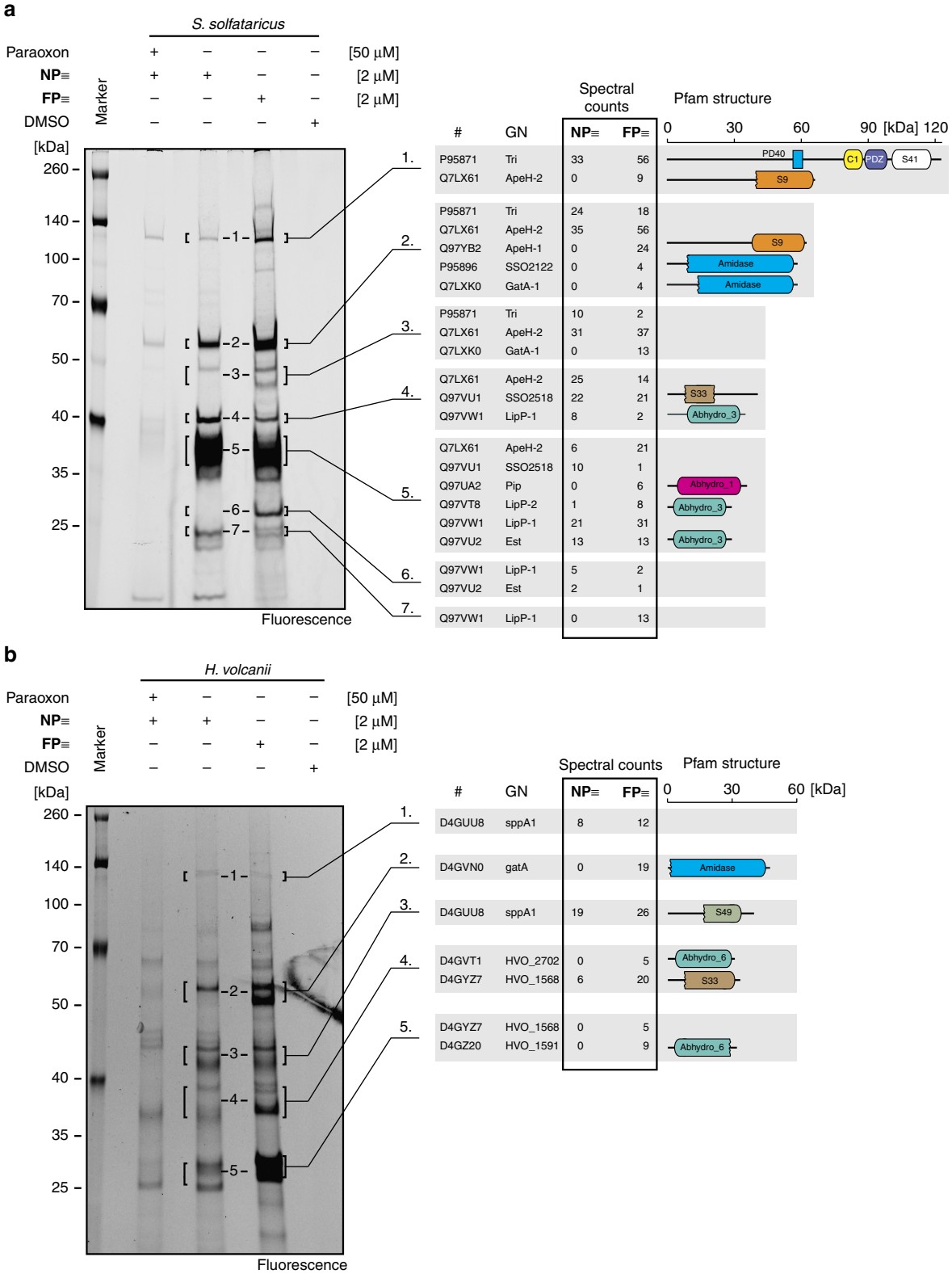

**Figure 3 | *In vivo* activity-based protein profiling of the thermoacidophile *Sulfolobus solfataricus* (P2) and the mesophilic halophile *Haloferax volcanii* (H26).** Large scale affinity purification and target identification of *in vivo* labelled *S. solfataricus* (**a**) and *H. volcanii* (**b**) after application of **FP≡** or **NP≡** with or without paraoxon pre-treatment (left panel) and corresponding Uniprot ID, gene name (GN), spectral counts and Pfam domain structure for identified serine hydrolases from *S. solfataricus* or *H. volcanii* (right panel). Please note that in some cases we did find evidence for high molecular weight proteins also in lower regions of the gel (indicated in the figure by multiple mentions of the corresponding proteins). This could have been caused by partial degradation, processing or insufficient gel separation of these large proteins.

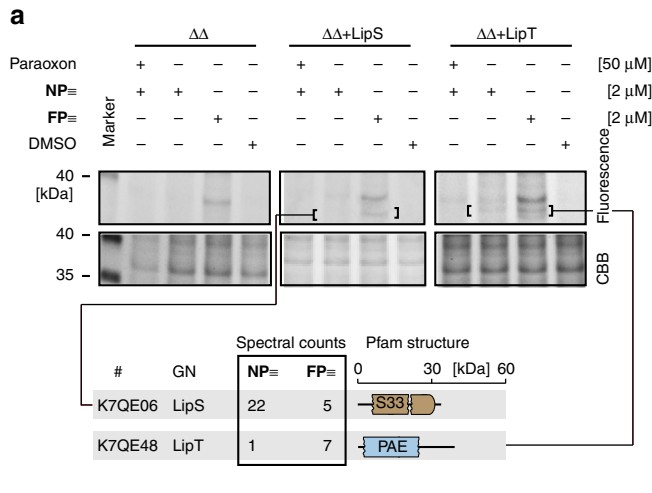

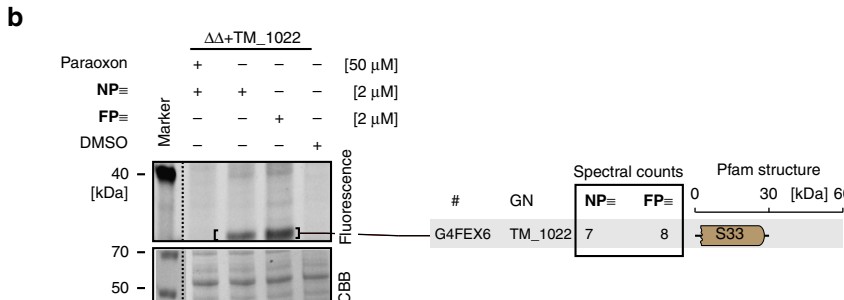

**Figure 4 | ABPP of heterologously expressed serine hydrolases in the *S. acidocaldarius* double deletion strain (ΔΔ).** 1D SDS–PAGE of the *in vivo* profiled ΔΔ *S. acidocaldarius* strain overexpressing metagenome-derived LipS and LipT via a host specific *mal* promotor (**a**) and TM_1022 from *T. maritima* via a native promotor (**b**) and corresponding Uniprot ID, gene name (GN), spectral counts and Pfam domain structure for identified serine hydrolases. Labelling was achieved with **FP**≡ or **NP**≡ with or without paraoxon pre-treatment.

(family PAE, pectinacetylesterase) were cloned under the control of the *Sulfolobus mal* promotor into the vector pSVAmZ-SH10. Both esterases are of bacterial origin and were previously isolated and characterized from a metagenome library[42]. Both proteins share little sequence homology with the lipases and esterase from *S. solfataricus* and *S. acidocaldarius* discussed so far. We then transformed the ΔΔ mutant with the two constructs and performed *in vivo* ABPP. The overexpression strains were incubated with 2 μM **NP**≡ after or without preincubation with 50 μM paraoxon and 2 μM **FP**≡. Fluorescence detection of proteins revealed an additional faintly labelled band either in the ΔΔ + LipS and the ΔΔ + LipT strain (Fig. 4a). Application of our affinity purification protocol and subsequent LC–MS-based target identification unambiguously established the identity of the two serine hydrolases. Thus, the *S. acidocaldarius* expression system using the host specific *mal* promotor is suitable for expression and ABPP-based detection of heterologously expressed serine hydrolases. Furthermore, the *S. acidocaldarius* esterase deletion strain (ΔΔ) is a suitable host strain for screening of heterologously expressed proteins.

Next, we investigated the utility of our approach for analysis of heterologously expressed enzymes from their native promotors. We selected the serine hydrolase TM_1022 (MW 28.8 kDa[41]) from the hyperthermophilic bacterium *Thermotoga maritima* because (i) we had access to a full genome sample from this organism and thus wanted to test if the bacterial promoter is recognized by the archaeal eukaryotic-like transcription machinery, (ii) its molecular weight should allow facile detection on the gel and (iii) TM_1022 features a S33 serine hydrolase domain that in principle can be robustly labelled[31].

Accordingly, we PCR-amplified the gene including a 500 bp upstream region of the transcription start (Supplementary Data 3) thereby ensuring that the native bacterial promoter is present in the amplified sequence and expressed it in the ΔΔ *S.acidocaldarius* double mutant strain. Subsequent *in vivo* ABPP labelling with 2 μM **NP**≡ after or without pre-incubation with 50 μM paraoxon and 2 μM **FP**≡ led to the appearance of one paraoxon-sensitive band with a molecular weight of ca. 30 kDa (Fig. 4b). Labelling was thereby more pronounced if **FP**≡ was used. The indicated gel region was excised from 1D SDS–PAGE gels and subjected to in-gel digestion with trypsin and subsequent target identification by LC–MS, resulting in a robust identification of TM_1022.

We were thus able to demonstrate expression and robust detection of a hyperthermophilic bacterial esterase from a genomic fragment under the control of the native bacterial promoter in the archaeal ΔΔ *S. acidocaldarius* esterase deletion strain. This indicates that *S. acidocaldarius* recognizes promoters even from distantly related organisms like the bacterium *T. maritima* (Fig. 1a). Thus, *S. acidocaldarius* qualifies as a host strain and in combination with the 'extremophilic' ABPP approach established herein is also applicable for the identification of (novel) serine hydrolases by functional screenings, for example, of metagenomic libraries of thermophilic origin.

**Enzyme activity assays validate MS results.** To assess the sensitivity to detect active enzymes by our *in vivo* ABPP approach and to validate the obtained findings, we performed

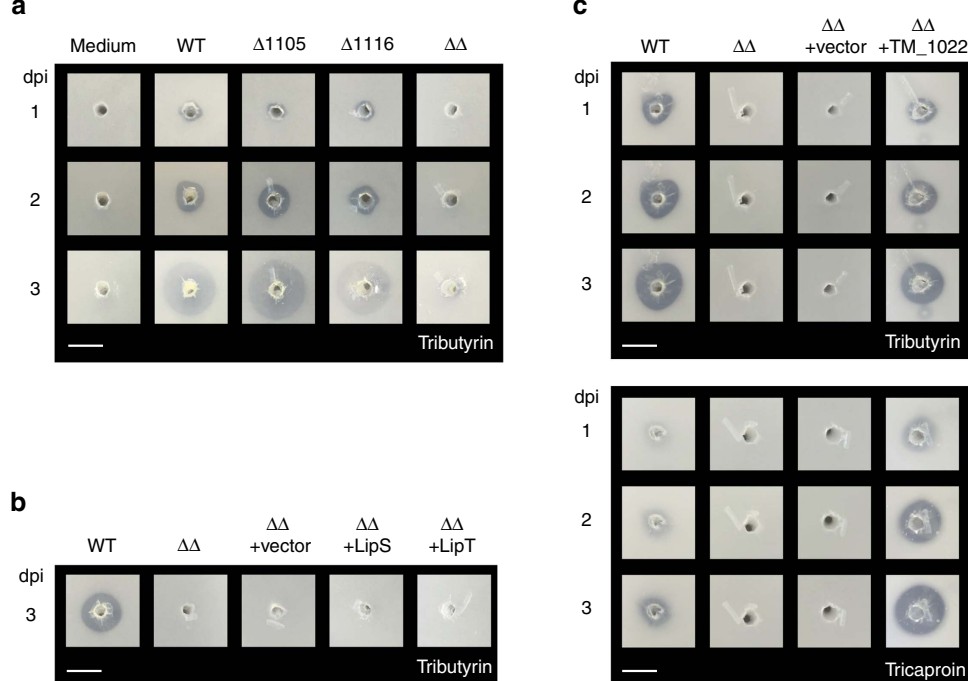

**Figure 5 | *In vivo* esterase activity assays using an esterase plate assay.** 20 µl of a culture with an $OD_{600}$ of 1 of the corresponding strains were spotted on plates containing the esterase substrates tributyrin or tricaproin and were incubated at 78 °C for the indicated number of days. Accordingly, esterase activity becomes visible by the appearance of a halo around the original infection site that indicates cleavage of the corresponding substrate. dpi: days post inoculation. (**a**) The esterase plate assay with the *S. acidocaldarius* MW001 (WT) and single knockout mutants Δ1105 and Δ1116 reveal esterase activity while the double knockout mutant ΔΔ is catalytically strongly impaired. (**b**) The esterase plate assay indicates no significant tributyrin-hydrolyzing activity for the ΔΔ mutant and the ΔΔ mutant transformed with the empty vector but also not for the ΔΔ mutant transformed with the metagenome-derived esterase LipS or LipT. (**c**) The ΔΔ mutant transformed with TM_1022 displays tributyrin, as well as tricaproin hydrolyzing activity while the ΔΔ mutant as well as the ΔΔ mutant transformed with an empty vector is inactive. White bar corresponds to 1 cm.

complementary enzyme assays with the WT, as well as the generated mutants or transformed strains.

As first experimental setup, we used a plate-based indicator assay in which a tributryin or tricaproin-containing agar was inoculated with the generated *S. acidocaldarius* cultures and incubated for up to three days at 78 °C. The addition of the different lipid substrates results in a turbidity of the agar that clears off upon hydrolysis by catalytically active esterases. Accordingly, the degradation of the substrates via active esterases results in the generation of a halo around the original inoculation site (Fig. 5). Larger halos thereby correspond to a higher overall esterase activity. This assay therefore represents a convenient and technically simple approach to functionally screen bacterial or archaeal cultures for esterase activity and similar setups are frequently used in functional *in vivo* screenings for novel microbial enzymes[43].

We started our enzyme activity assays with *S. acidocaldarius* cultures with *S. acidocaldarius* MW001 (WT) strain, both single knockout strains Δ1105 and Δ1116, as well as the double deletion strain ΔΔ. Previous studies with inoculation of a tributyrin-containing agar followed by growth of these cultures for up to three days led to the time-dependent formation of a halo for the WT, as well as Δ1105 and Δ1116 mutants but not for the ΔΔ mutant (Fig. 5a). Thus, in accordance with our ABPP experiments, the WT and the single knockout mutants displayed esterase activity, while the ΔΔ mutant was inactive. Importantly, a control inoculation with only medium did not induce any tributyrin hydrolysis.

We then continued our enzyme assays with the heterologously expressed proteins. To this end, we first analysed the *S. acidocaldarius* ΔΔ mutant overexpressing either LipS or LipT

(ΔΔ + LipS and ΔΔ + LipT, Fig. 5b) Even after three days, no significant hydrolysis could be observed. Of note, LipS and LipT were previously reported as triacylglycerol lipases that, despite a substrate preference for C8 and C10 fatty acids, were reported to display activity also versus C4 substrates[42]. This indicates that the generated LipS and LipT protein levels are insufficient to generate a reliable readout in this assay conditions and emphasizes the sensitivity advantage of the ABPP-based enzyme detection. As expected, the corresponding controls, consisting of the WT strain that displayed enzymatic activity, and the ΔΔ mutant, as well as a ΔΔ mutant with a transformation vector lacking the corresponding inserts did not show substrate hydrolysis.

Finally, we also analysed the TM_1022 expressing ΔΔ mutant (Fig. 5c). As this enzyme is biochemically uncharacterized so far, we used two different substrates in our assay, that is tributyrin and tricaproin. For both substrates, a time-dependent formation of a halo could be observed for this culture; the halo for the tricaproin substrate was thereby slightly larger than for tributyrate, indicating that TM_1022 displays a substrate preference for longer C chain fatty acids. The corresponding controls, that is an enzymatically active WT and an inactive ΔΔ mutant, as well as an inactive ΔΔ mutant transformed with an empty vector, delivered the expected results. Thus, our plate assay was able to confirm the results from the ABPP measurements for this strain.

To corroborate our findings from the plate assays, we then performed a secondary biochemical esterase activity assay with the generated cultures. In this assay, lysates of the cultures were prepared and tested for their capability to hydrolyse the chromophoric *para*-nitrophenol octanoate ester (**C8-pNP**). Accordingly, we generated lysates of the ΔΔ mutant, the Δ1105

and Δ1116 mutants, as well as of the ΔΔ mutant expressing either LipS, LipT or TM_1022 and determined their overall esterase activity via monitoring of *para*-nitrophenol liberation from **C8-pNP** either with or without preincubation with 250 μM paraoxon (Supplementary Fig. 6). As expected from our previous results, the ΔΔ mutant thereby did not display any enzymatic activity, while the Δ1105 and Δ1116 lysates were still highly active. In contrast, the LipT and TM_1022 containing lysates displayed only a weak enzymatic activity and the LipS lysate was basically inactive, although for both LipS and LipT activity with **C8-pNP** was reported[42]. In accordance with the ABPP results, pre-incubation with paraoxon led in all cases to a significantly reduced esterase activity. Also, pre-incubation with ABPs **NP≡** and **FP≡** (2 μM each) resulted in a nearly complete loss of esterase activity in the ΔΔ mutant expressing the Saci_1105 (Supplementary Fig. 7). The **NP≡** concentration dependent exponential activity decay was determined and revealed an $IC_{50}$ value of 0.15 μM indicating high labelling efficiency of the ABP.

Overall, our enzyme assays confirm the results from the ABPP experiments. In some cases, the ABPP approach however turned out to be much more sensitive as alternative enzyme assays failed to detect the corresponding esterase activity.

## Discussion

We have developed a robust method for *in vitro* and *in vivo* ABPP detection of active enzymes, either native or heterologously expressed, applicable under harsh environmental conditions, that such as, high temperatures, low pH and high salinity, respectively. We also demonstrated that ABPP is compatible with archaeal organisms which often thrive under such harsh conditions but also differ in many respects like metabolism and enzyme repertoire as well as cell wall and membrane structure from bacteria and eukarya[13]. So far, only one approach of ABPP in Archaea was reported; this study was, however, performed with lysates under non-physiological conditions and lacked a thorough identification of labelled proteins[24]. Our general ABPP workflow in Archaea with this broad applicability concerning physical and environmental conditions does not only allow to experimentally classify enzymes and to confirm their affiliation to enzyme families/classes but also to experimentally demonstrate that a protein is indeed expressed and catalytically active[23]. The ABPP approach thus enables a very rapid characterization of the active serine hydrolase repertoire of archaeal organisms which with standard molecular biology and biochemistry techniques is much more labour-intensive, time consuming and difficult to achieve. Notably, the here described ABPP methodology has a broad application potential and also allows use of other enzyme class/family-specific ABPs[16,18]. As chemical biologists constantly develop novel ABPs[19], many more enzyme classes should therefore be addressable by our approach.

Moreover, we were able to demonstrate the high sensitivity of our approach as we were able to detect and to profile enzyme activities *in vivo* (for example, for LipT) with one methodological setup that we could not directly measure with other established methodologies for functional screenings such as plate assays or measurement of enzymatic activities in lysates. The lack of enzymatic activity in these assays may be caused by low expression levels, weak biochemical stability or deactivation in the lysate conditions or the use of an improper substrate; consequently, it is reasonable to assume that a careful and time-consuming optimization of assay conditions might lead to a successful detection of these enzymes in the established assay systems. In any case, these approaches however stand in contrast to the robust and technically simple identification of these enzymes in the ABPP experiment, thus highlighting that

a reliable, straightforward and sensitive detection of active enzymes via ABPP does not require a previous optimization of assay conditions.

For the detection of active enzymes, we used a gel-based fluorophore approach, which is generally considered as the most rapid and cost economic ABPP approach. Furthermore, as shown here, the genetic system available for *S. acidocaldarius* (as well as in principle for some other archaea) allows for the construction of a suitable 'low background' strain that allows functional gene screening for low abundant or weakly active enzymes. Of note, for low abundant or weakly active native or heterologously expressed proteins, a more sensitive detection of active enzymes is meanwhile also achievable by a gel-free ABPP approach[16,45]. We expect that our generalized workflow will also be compatible with such a modified protocol since it only requires a different MS-compatible sample preparation after Click chemistry labelling.

We therefore believe that our established ABPP workflow is not only of interest for biological and biochemical investigations, where it may find application for example to unravel the metabolic potential of an organism or to follow *in vivo* activity profiles of different protein classes under different growth conditions (for example, different carbon sources), but is also a valuable new method for functional genomics approaches under extremophilic conditions in general and particularly in living archaea. We anticipate that it may also find applications in modern biotechnology, for example, for identifying heat resistant enzymes for industrial applications by a direct investigation of environmental samples with combination of metagenomics approaches.

## Methods

**Cultivation of archaeal and and bacterial strains.** *Sulfolobus acidocaldarius* MW001, a Δ*pyrEF* strain, the *S. acidocaldarius* esterase knockout mutants MW901 (MW001Δ1105), MW902 (MW001Δ1116) and MW903 (MW001Δ1105Δ1116) and plasmid-containing strains as well as *S. solfataricus* were grown aerobically in Brock medium at a pH of 3 and 78 °C. The medium was supplemented with 0.1% (w/v) tryptone or with 0.1% (w/v) N-Z-Amine. The growth medium for MW001 and the deletion mutants was supplemented with 10 μg ml$^{-1}$ uracil. For the esterase expression experiments, no uracil was added but expression was induced by addition of 0.4% (w/v) dextrin. The growth of the cells was monitored by measurement of the optical density at 600 nm ($OD_{600}$).

*H. volcanii* strain H26 (pyrE2 knock-out strain) was grown at 45 °C in sterile rich Hv-YPC media according to Allers et al.[44].

*E. coli* ER1821 cells were used to obtain methylated plasmids at all HaeIII sites as reported previously[46]. For cloning, the *E. coli* K-12 DH5α (DSMZ 6897) strain and for expression of recombinant proteins an *E. coli* Rosetta (DE3) (Novagen), harbouring the plasmid pRARE (CamR), was used.

**Construction of Archaea deletion plasmids.** For markerless deletion of the two esterases encoding genes S*aci_1105* and S*aci_1116*, 500 bp of the 3′ end of each coding region were amplified by PCR. The purified PCR products were cloned into the ApaI and XhoI restriction site of pSVA431 (ref. 46), resulting in the plasmids pSVA485 and pSVA487, respectively. The upstream and downstream region with a size of around 500 bp were amplified by PCR. The PCR products were purified and an overlap extension PCR was performed. The purified overlap extension PCR products were cloned into the second MCS of the deletion pre-vectors pSVA485 and pSVA487 using AvrII and EagI to give the final vectors pSVA486 and pSVA488, respectively. All used deletion (del) primers are reported in the primer list (Supplementary Data 3).

**Constructions of archaeal mutant strains.** Linear markerless deletion fragments were obtained by PCR using the plasmids pSVA486 or pSVA488 as template and the corresponding gene forward and downstream reverse primer. The resulting PCR products harboured next to the gene deletion specific regions of the *Sulfolobus solfataricus pyrEF* marker cassette and the *S. solfataricus lacS* reporter cassette. Approximately 50 μl competent MW001 cells with a theoretical $OD_{600}$ of 10 were electroporated with 100 ng of the purified PCR product using a GenPulser MX cell (Biorad) (2,000 V, 660 Ω, 25 μF, 1 mm cuvette)[46]. Cells were resuspended in 450 μl minimal Brock medium (pH 5) and recovered for 30 min at 75 °C while moderately shaking. 100 μl recovered cell mixture was spread on a first selection plate (Brock medium supplemented with 0.1% (w/v) N-Z-Amine, 0.2% (w/v) dextrin,

0.7% (w/v) gelrite) and incubated in a closed plastic box at 75 °C for 5–7 days. Colonies were sprayed with 5 mg ml$^{-1}$ X-Gal solution and the blue integrants were transferred to a second selection plate (Brock medium supplemented with 0.1% (w/v) N-Z-Amine, 0.2% (w/v) dextrin, 10 µg ml$^{-1}$ uracil, 100 µg ml$^{-1}$ 5-fluoroorotic acid (5-FOA), 0.7% (w/v) gelrite). The second selection plate was incubated in a closed plastic box at 75 °C for 5 days. The white colonies after X-Gal spraying (see above) were screened by colony PCR for successful gene deletion. Gene deletion was additionally confirmed by sequencing with primers binding outside of the cloning region (Supplementary Data 3).

**Construction of expression plasmids pSVA1551 and pSVAmZ-SH10.** To optimize cloning into the *S. acidocaldarius* expression vector, the *Bam*HI restriction sites were removed from the original expression vector pSVA1431 (ref. 38). To this end and to reduce the size of the pRN1 backbone, five backbone fragments with overlapping ends were amplified via PCR using the 'mod_*Bam*HI del' primers. The purified PCR products were assembled by overlap PCR reactions to one fragment, which was cloned into pSVA1431 by restriction with *Eag*I and *Xma*I. In the resulting plasmid another 36 bp were deleted by amplification of the whole plasmid using the primers mod_MCS_del_fw and mod_MCS_del_rev. The purified PCR product contained 22 overlapping base pairs at both ends and was transformed into *E. coli* DH5α, which led to the recircularization of the plasmid. Next, a modified multiple cloning cassette was amplified from the previous prevector pMZ1 with the primers mod_MCS_ins_fw and mod_MCS_ins_fw and cloned via restriction with *Nco*I and *Eag*I into the recirculized plasmid. All modifications were verified by sequencing.

For cloning of pSVAmZ-SH10 the minimal replicon of pRN1 consisting of the region surrounding orf56 and orf904 was shortened and the remaining NdeI site simultaneously removed[47]. To this end, two backbone fragments were amplified using the primer combinations mod_Size_fw, mod_NdeI_del_rev and mod_NdeI_del_fw, mod_Size_rev and assembled by overlap PCR. The full-length product was purified and cloned into pSVA1551 by restriction with *Not*I and *Xma*I. To allow blue/white screening, a lacZ fragment with an attached multiple restriction site was amplified from *E. coli* K12 genomic DNA by PCR with the primers mod_lacZ_fw and mod_lacZ_rev and cloned into the previous plasmid by insertion into the NcoI and BamHI restriction sites. The plasmid modifications were verified by sequencing.

**Construction of archaeal esterase overexpression strains.** In this study, different esterases were expressed in *S. acidocaldarius*: TM_1022 from *Thermotoga maritima* MSB8, Saci_1105 and Saci_1116 (ref. 48) from *S. acidocaldarius* and the two thermophilic esterases LipT and LipS from a metagenome sample[42]. These esterase genes were amplified by PCR; for *lipS* and *saci_1116* an internal NcoI restriction site was removed by overlap extension PCR introducing a silent mutation in the respective position. The gene *tm1022* was cloned with its own promoter, that is, including an approximately 500 bp region upstream of its start codon. The purified PCR products were restricted by the endonucleases depicted in Supplementary Data 3 and ligated into the vectors pSVAmZ-SH10, pSVA1551 or pSVA1431 treated with the same enzymes and alkaline phosphatase. The plasmids were amplified in *E. coli* DH5α. The following final constructs were used for this study: pSVA-saci_1116, pSVAmZ-saci_1105 pSVAmZ_lipS, pSVAmZ_lipT and pSVA-p-tm1022 (Supplementary Data 3).

Successful cloning was confirmed by sequencing, and the plasmids were transformed into *E. coli* R1821 cells for methylation. Electrocompetent *S. acidocaldarius* cells (50 µl, theoretical OD$_{600}$ of 10) were electroporated with 400 ng of methylated recombinant plasmid (1,500 V, 600 O, 25 µF; 1-mm cuvette) as previously reported[49]. The cells were resuspended with 50 µl 2 × recovery solution (1% sucrose, 20 mM β-alanine, 1.5 mM malate buffer (pH 4.5), 10 mM MgSO$_4$) and incubated for 30 min at 75 °C in a Thermomixer. Cells were mixed with 800 µl Brock medium and spread on 1st-selection plates (Brock medium supplemented with 0.1% N-Z-Amine, 0.2% dextrin, and 0.7% Gelrite) by using on a vertical shaker. The plates were sealed in a plastic bag and were incubated at 78 °C for 5 to 7 days. Single-colonies were picked and were transferred to liquid first-selection medium and single-colonies were screened for the inserted gene by colony PCR using the plasmid-based flanking primers pSVA-F: CGGAGGTGTCCTTAA GTTTAG and pSVA-R: CGGGCGTGATAAAGTCTGTCTC)[49].

A 7-ml preculture was subsequently grown for 3 days in Brock medium containing 0.1% tryptone and 0.2% dextrin to initiate expression under the control of the mal-promotor. For the ABPP experiments, 50 ml of the same media was inoculated with preculture (2% (v/v)) and grown to an OD$_{600}$ of 0.8–1.

**Two-step ABPP for detecting archaeal serine hydrolases.** All probes or inhibitors were dissolved in DMSO. Labelling of active serine hydrolases was performed in 3.5 ml liquid culture with an OD$_{600}$ of 1.0. Competitive ABPP was performed by pre-incubation with 50 µM paraoxon or 50 µM PMSF for 10 min at 78 °C for *S. acidocaldarius* and *S. solfataricus* and 45 °C for *H. volcanii*. 2 µM **NP≡** or 2 µM **FP≡** were added and the resulting mixture was incubated for 2 h under cultivating conditions. The cells from 3 ml culture were harvested and the pellet washed with culture medium or in some cases with 2% SDS. The cells were subsequently lysed by addition of 50 mM HNa$_2$PO$_4$ pH 8.0 buffer (200 µl) and

sonication with a biodisruptor (Diagenode) using the following conditions: 1 minute pulse and 30 s pause in 5–7 cycles with high power. The extracts were then cleared by centrifugation (15,000 r.p.m., 4 °C, 5 min) to separate soluble proteins from cell debris. The protein concentration of the resulting proteome lysate was determined by a Bradford assay. For large scale labelling we used 1 mg total protein and for small scale labelling 50 µg. To realize the 'Click' Huisgen-type Cu(I)-catalysed $(3+2)$ cycloaddition the protein solution was mixed with 10 µM of the click-reporter (either **Rh-N$_3$** or **Biot-Rh-N$_3$**), 100 µM Tris[(1-benzyl-1*H*-1,2,3-triazol-4-yl)methyl]amin (TBTA) and 1 mM Tris(2-carboxyethyl)phosphin (TCEP). Finally, to start the reaction, 1 mM CuSO$_4$ was added and the resulting mixture was incubated for 1 h in the dark at room temperature.

**Two-step ABPP and identification of archaeal serine hydrolases.** Before affinity purification, unbound reporter and salts were removed by passing the cell lysate after the click chemistry reaction through a PD-10 size exclusion column[50]. The resulting protein lysates were incubated with avidin beads (Sigma) for 1 h while gently rotating at room temperature in the dark. The beads were washed at least 5 × with 1% SDS in PBS buffer (pH 7.4) before adding 1 equivalent 2 × LDS gel loading dye (53 mM Tris HCl; 70 mM Tris base; 1% lithium dodecyl sulfate (LDS); 5% glycerol; 0.25 mM EDTA; 0.11 mM SERVA Blue G250; pH 8.5; final 1 × LDS buffer). Samples were incubated for 10 min at 90 °C to release avidin-bound proteins. Separation of released proteins was then achieved by gel electrophoresis and labelled proteins were visualized by scanning fluorescence on a Typhoon FLA 9,000 scanner (GE Healthcare). Labelled proteins were excised and the gel pieces were washed 2 × with water and 2 × with 100 mM ammonium bicarbonate solution. The proteins were reduced with 10 mM TCEP (30 min, 62 °C) while gently shaking and subsequently alkylated by adding 55 mM iodoacetamide (IAA) (30 min, room temperature, in the dark). The gel slabs were washed three times with acetonitrile (3 × 50 µl). After the last wash, the gel slabs were completely dried by using a vacuum concentrator (Eppendorf) for 5 min. The dried gel pieces were incubated with a 10 ng µl$^{-1}$ trypsin solution (200 µl, 37 °C, 16 h, vigorous shaking). The digestion was stopped by addition of formic acid to a final concentration of 5% (v/v) formic acid. The supernatant was transferred to a fresh Eppendorf tube. The remaining gel pieces were subsequently washed three times with acetonitrile (3 × 50 µl). The supernatants of these washes were combined with the recovered digestion mix and dried in a vacuum concentrator (Eppendorf).

**Sample clean-up for LC–MS.** The dried peptide pellets were dissolved in 0.1% formic acid (FA) solution and desalted on home-made C18 StageTips[51]. The tryptic digests were passed over a two disc StageTip and the immobilized peptides washed with 0.1% FA. The peptides were eluted from the StageTips with 80% ACN 0.5% FA. After elution from the StageTips, samples were dried using a vacuum concentrator (Eppendorf) and the peptides were resuspended in 0.1% (v/v) formic acid solution (10 µl).

**LC–MS/MS.** LC–MS/MS Experiments were performed on an Orbitrap Elite instrument (Thermo) that was coupled to an EASY-nLC 1000 liquid chromatography (LC) system (Thermo). The LC was operated in the one-column mode. The analytical column was a fused silica capillary (75 µm × 20–30 cm) with an integrated PicoFrit emitter (New Objective) packed in-house with Reprosil-Pur 120 C18-AQ 1.9 µm resin (Dr Maisch). The analytical column was encased by a column oven (Sonation) and attached to a nanospray flex ion source (Thermo). The column oven temperature was adjusted to 45 °C during data acquisition and in all other modi at 30 °C. The LC was equipped with two mobile phases: solvent A (0.1% formic acid, FA, in water) and solvent B (0.1% FA in acetonitrile, ACN). All solvents were of UPLC grade (Sigma). Peptides were directly loaded onto the analytical column with a maximum flow rate that would not exceed the set pressure limit of 980 bar (usually around 0.8–1.0 µl min$^{-1}$). Peptides were subsequently separated on the analytical column by typically running a 50 min gradient of solvent A and solvent B (start with 7% B; gradient 7 to 35% B for 40 min; gradient 35–100% B for 5 min and 100% B for 5 min) at a flow rate of 300 nl min$^{-1}$. The mass spectrometer was operated using Xcalibur software (version 2.2 SP1.48) and was set in the positive ion mode. Precursor ion scanning was performed in the Orbitrap analyser (FTMS) in the scan range of *m/z* 300–1,500 and at a resolution of 60,000 with the internal lock mass option turned on (lock mass was 445.120025 *m/z*, polysiloxane)[52]. Product ion spectra were recorded in a data dependent fashion in the ion trap (ITMS) in a variable scan range and at a rapid scan rate. The ionization potential (spray voltage) was set to 1.8 kV. Peptides were analysed using a repeating cycle consisting of a full precursor ion scan (1.0 × 10$^6$ ions or 50 ms) followed by 12 product ion scans (1.0 × 10$^4$ ions or 100 ms), where peptides are isolated based on their intensity in the full survey scan (threshold of 500 counts) for tandem mass spectrum (MS2) generation that permits peptide sequencing and identification. CID collision energy was set to 35% for the generation of MS2 spectra. During MS2 data acquisition dynamic ion exclusion was set to 120 s with a maximum list of excluded ions consisting of 500 members and a repeat count of one. Ion injection time prediction, preview mode for the FTMS, monoisotopic precursor selection and charge state screening

were enabled. Only charge states higher than one were considered for fragmentation.

**Peptide and Protein Identification using MaxQuant.** RAW spectra were submitted to an Andromeda[35] search in MaxQuant (version 1.5.3.30) using the default settings[53]. Label-free quantification and match-between-runs was activated. MS/MS spectra data were searched against the respective reference databases downloaded from the Uniprot website (*Sulfolobus acidocaldarius* (TaxID 330779): UP000001018_330779.fasta (2,221 entries); *Haloferax volcanii DS2* (TaxID 309800): UP000008243_309800.fasta (3987 entries); *Sulfolobus solfataricus P2* (TaxID 273057): UP000001974_273057.fasta (2938 entries); *Thermotoga maritima* (TaxID 243274): UP000008183_243274.fasta (1852 entries)). All searches included a contaminants database (as implemented in MaxQuant, 267 sequences). The contaminants database contains known MS contaminants and was included to estimate the level of contamination. Andromeda searches allowed oxidation of methionine residues (16 Da) and acetylation of the protein *N*-terminus (42 Da) as dynamic modifications and the static modification of cysteine (57 Da, alkylation with iodoacetamide). Enzyme specificity was set to 'Trypsin/P'. The instrument type in Andromeda searches was set to Orbitrap and the precursor mass tolerance was set to $\pm 20$ p.p.m. (first search) and $\pm 4.5$ p.p.m. (main search). The MS/MS match tolerance was set to $\pm 0.5$ Da. The peptide spectrum match FDR and the protein FDR were set to 0.01 (based on target-decoy approach). Minimum peptide length was seven amino acids. For protein quantification unique and razor peptides were allowed. Modified peptides were allowed for quantification. The minimum score for modified peptides was 40.

**Tributyrin and tricaproin indicator plate assay.** For detection of esterase activity, whole cells were spotted on tributyrin or tricaproin containing indicator plates. For the preparation of the indicator plates, 1.4% (w/v) gelrite was solved in 0.5 l hot water. The resulting gelrite solution was mixed with 0.5 l $2\times$ Brock medium (supplemented with 20 mM $MgCl_2 \times 6H_2O$, 6 mM $CaCl_2$, 0.2% (w/v) N-Z-Amine, 0.4% (w/v) dextrin, without uracil) and 1% (v/v) tributyrin or tricaproin were applied. The resulting lipid suspension was homogenized for 5 s by a disperser (IKA, Staufen) and the plates were poured. After solidification of the medium, cavities were punched into the plates using a 5 ml pipette tip. 20 µl of sample cultures pre-grown to an $OD_{600}$ of 1 in liquid Brock media with 0.1% (w/v) N-Z-Amine were applied into the cavities and incubated for up to 3 days at 78 °C. The formation of halos by triacylglycerol and tricaproin degradation was followed visually.

**Biochemical esterase activity assay.** For detection of esterase activity in *S. acidocaldarius* crude extracts, 100 ml expression cultures were grown and harvested under the above described conditions. Cell pellets (0.5–2.0 g) were resuspended in a three-fold buffer volume (50 mM $NaH_2PO_4$, pH 8.0) and disrupted by using the Precellys 24 system (0.1 mm beats, 6,500 r.p.m. $3 \times 20$ s, Bertini technologies). Cell debris was removed by centrifugation at 21,000*g*, 45 min, 4 °C. For detection of esterase activity 20–600 µg crude extract were applied in photometric activity tests (405 nm) using 0.5 mM pNP-octanoate at 35 °C as a substrate. For esterase inhibition assays, the enzyme was pre-incubated in the presence of 250 µM paraoxon (1 h at 60 °C) before the assay was started by addition of pNP-octanoate.

**Data availability.** All relevant data and the Supplementary Materials are available from the corresponding authors on request.

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

## Acknowledgements

We thank Prof Dr Wolfgang Streit (University Hamburg) for providing the LipS/LipT metagenome derived clones. Funding of the present study by an ERC Starting Grant (grant no. 258413, to M.K.), the DFG (grant no. INST 20876/127-1 FUGG to M.K.), the European Union (7th framework programme for research and technological development (FP7), HotZyme (Project reference: 265933) to B.S.), the BMBF (ExpresSYS, grant No. 0315586C to B.S. and grant no. 0315586D to S.V.A.; SulfoSYSBiotec, grant no. 0316188C to S.V.A.) and by the Ministerium für Innovation, Wissenschaft und Forschung des Landes Nordrhein-Westfalen, the Senatsverwaltung für Wirtschaft, Technologie und Forschung des Landes Berlin and the Bundesministerium für Bildung und Forschung (to A.S.) is gratefully acknowledged.

## Author contributions

M.K., F.K., C.B. and B.S. conceived and designed the study; S.Z., V.K., S.N., S.N., J.V., M.B., A.W. and S.-V.A. performed and analysed all chemical biology experiments. I.F., A.S. and F.K. performed mass spectrometry and related data analysis. M.K. and B.S. supervised the project and wrote the manuscript.

## Additional information

**Competing interests:** The authors declare no competing financial interests.

