## [Peer Review File · Nature Communications]

Reviewers' comments:

Reviewer #1 (Remarks to the Author):

Dear Editor and Authors,

It was with great pleasure that I reviewed the paper titled 'activity based protein profiling as a robust method for enzyme identification and screening in extremophilic Archaea'. In this manuscript, Zweerink et al. describe the in vivo ABPP of thermophilic hydrolases and in doing so they really push the boundary of the field to the extreme. Moreover, they have done this in a thorough and complete manner, which leaves me with only minor experimental points to address (as well as some textual ones).

Before I continue with this review, I do wish to point out that ABPP is my field of interest, which meant that I could not rightfully comment on technical detail about other aspects of the paper, such as the recombinant cross-species expression of enzymes in their *Sulfolobus* host.

Some points on the paper that I would like to see clarified are the following:

- 1) Are the fluorophosphonate and pNP-phosphonate equally stable under the culture conditions? And could this go some way in explaining the better labeling of the pNP-phosphonate?
- 2) Can the increased temperatures also increase background binding? This is largely addressed in the supplementary information, but I think that a lysate labeling of e.g. *E. coli* lysate at both room temperature and 78 degrees would be useful to determine the extent of this increase in background.
- 3) One experiment that, if possible, I thought would be a nice way to determine the efficiency of the ABPP is to perform the esterase activity experiments described on page 15 and 16 in presence of the ABP to determine what percentage of overall esterase activity is blocked by the ABPP.

These are my minor suggestions that I think would make this an even nicer manuscript. Textually, I think the manuscript would benefit from the removal of some rather long sentences, such as Page 4, lines 49-53, 54-56, page 5 86-89.

In conclusion, I think this is an exciting paper that is very suitable for Nature Communications, as it presents a new application of in vivo ABPP to a very difficult class of enzymes.

Reviewer #2 (Remarks to the Author):

The manuscript, Activity based protein profiling as a robust method for enzyme identification and screening in extremophilic Archaea, reports the development/demonstration of a robust and sensitive method to directly identify active serine hydrolases in extremophile cells in vivo. The method successfully identified serine hydrolases from thermoacidophilic archaea (*Sulfolobus*) and a halophile (*Haloferax*). The authors demonstrate that by combining the ABPP system with the genetic manipulation tools developed for *Sulfolobus acidocaldarius*, it is possible to directly screen for thermostable serine hydrolases that may be encoded in metagenome libraries. This was shown with two genes from a bacterial metagenome library and a serine hydrolase gene from a thermoneutrophilic bacterium (*Thermotoga*). The versatility of the method is clearly shown and should attract the broad attention of those engaged in microbiology, metabolism, enzyme screening, biotechnology and chemical biology. Furthermore, the study should encourage the development of similar systems using host cells that exhibit different extremophilic properties.

Genetic systems are available for a variety of halophiles (Halobacterium, Haloferax), thermoneutrophilic archaea (Pyrococcus, Thermococcus), Psychrophiles (Shewanella), Alkaliphiles (Alkaliphilic Bacillus) and piezophiles (Thermococcus barophilus), which have the potential to be utilized as host cells for in vivo screening.

The applicability and versatility of the method is clearly proven with numerous experiments. Experiments were carefully designed and technically sound. The manuscript is well written and was easy to read. Specific comments are as follows. Comments 1, 10, 11, 15, 18, 19, 20 are relatively more important than the others.

1. Abstract: the first two sentences can be combined and shortened so that the abstract sheds more light on the results of the manuscript.
2. Abstract last line: Is the term archaeal necessary here?
3. Line 119 constantly being developed?
4. Figure 1: define RG
5. Line 94: Spell out Sulfolobus
6. Lines 125-127: no need to parenthesize
7. Line 153: grows optimally?
8. The title of the manuscript shown in the Supplementary Information section differs to that of the main text. Please unify.
9. There are only 6 supplementary figures, not 8. Revise or remove "Content".
10. Supplementary Figure 2: Is the low molecular weight band in the Hp-NP lane relevant? It seems that it is not labeled at high temperature.
11. Line 193: Can the authors describe the term broadest in more detail? Does this mean the probe that labels the highest number of proteins, or are the authors subtracting Paraoxon plus from Paraoxon minus? Also please rephrase gold standard.
12. It is very difficult to read the domain labels on the right panel of Supplementary Figure 3. Please enlarge (the letters) or maybe just indicate the gene annotations.
13. Supplementary Figure 4, perhaps the molecular size markers should be indicated separately for the left and right gels?
14. Line 262: period
15. The reviewer had difficulty understanding which data sheet corresponded to the Supplementary Tables.
16. Line 275: Please write *S. acidocaldarius* before 1105.
17. Is it correct that SSO2517, or EST, was labeled?
18. Overall, the results seem to suggest that labeling intensity correlates well with total activity. Does this also apply for labeling efficiency? The reviewer could not really conclude on this. Is a more active protein labeled more, are the differences in labeling intensity due to the presence of more protein? If possible, this should be discussed.

19. Is the incubation time with NP= or FP= (2 h) an optimized time period? Is the effect of incubation time very large?

20. Does the absence of 1105 and 1116 result in the enhanced labeling of any other proteins? Or is the concentration of probe already saturated under the applied conditions?

Reviewer #3 (Remarks to the Author):

The paper entitled "ABBP-based detection of native and heterologously expressed extremophile enzymes in Archaea" tries to use the "activity based protein profiling" to identify enzymes, more specifically serine hydrolases, and introduce it as a screening method for enzymes from extremophile microorganisms, in particular Archaea.

The authors have applied this method both in vitro and in vivo to identify serine hydrolases. In vitro they have used recombinant proteins and the results have been successful with esterases. This method or technique can provide researchers working with archaea with a chemical tool to set out the assignment of functions to many hypothetical enzymes that find in archaeal genomes. The work is well done and perfectly detailed to reproduce by others researchers.

The approach is very interesting but the most important concern about the applicability is the developing of novel ABPs to identify more enzyme classes.

The paper deserves to be published.

Point-by-point response to reviewer comments

Reviewer #1 (Remarks to the Author):

Dear Editor and Authors,

It was with great pleasure that I reviewed the paper titled 'activity based protein profiling as a robust method for enzyme identification and screening in extremophilic Archaea'. In this manuscript, Zweerink et al. describe the in vivo ABPP of thermophilic hydrolases and in doing so they really push the boundary of the field to the extreme. Moreover, they have done this in a thorough and complete manner, which leaves me with only minor experimental points to address (as well as some textual ones).

Thank you very much for your very positive feedback on our manuscript.

Before I continue with this review, I do wish to point out that ABPP is my field of interest, which meant that I could not rightfully comment on technical detail about other aspects of the paper, such as the recombinant cross-species expression of enzymes in their *Sulfolobus* host.

Some points on the paper that I would like to see clarified are the following:

1) Are the fluorophosphonate and pNP-phosphonate equally stable under the culture conditions? And could this go some way in explaining the better labeling of the pNP-phosphonate?

To test the stability of both probes under culture conditions, we added both of them to culture medium and incubated the resulting mixture for 0 min, 15 min, 30 min, 60 min and 120 min at 78 °C, followed by analysis *via* LC-MS. Although the fluorophosphonate (**FP**) probe turned out to be less stable than the **NP**-probe, we could robustly detect its presence even after 120 min (and thus the incubation time used during the experiments).

We are however uncertain which conclusions can be drawn from this, in particular if this finding could explain a 'better' labelling of the **NP** probe. According to the gel analysis in Fig. 2, both probes basically led to an equally intense band at 38 kDa (corresponding to the acetyl esterase Saci_1116 and esterase/lipase Saci_1105); we therefore feel uncomfortable to state that **NP** labels these enzymes more efficiently (or alternatively 'better') than **FP**. At least in this case, the stability of **FP** vs. **NP** does not seem to be relevant for labelling efficiency.

FP nevertheless labels more proteins than **NP** and **NP** is therefore more specific (which could be considered as 'better' labelling). This enhanced labelling efficiency of **FP** vs **NP** has however been recognized before in other organisms (see for example *Bioorg Med Chem* 2012, 20, 601 for a study in *Arabidopsis*) and seems to be a consequence of the higher serine hydrolase reactivity of **FP** vs. **NP** and not due to different chemical stability in the medium (indeed, a lower **FP** stability should lead to less labelling, we however see the opposite effect).

In light of this data, we prefer to not discuss a correlation between probe stability and labelling efficiency in our study as we are not really able to make valid conclusions about this topic.

2) Can the increased temperatures also increase background binding? This is largely addressed in the supplementary information, but I think that a lysate labeling of e.g. *E. coli* lysate at both room temperature and 78 degrees would be useful to determine the extent of this increase in background.

As requested, we labelled *E. coli* lysates at 25 °C and 78 °C with **FP** and **NP** (and DMSO as a control).

At 25 °C, we observed only weak labelling if **NP** was used, while **FP** resulted in one prominent band at ca. 34 kDa and several other more-or-less faintly labelled bands. No band (including the 34 kDa band visible with **FP**) was competed by pre-incubation with paraoxon.

At 78 °C, significant more labelling became visible with **NP** and **FP**. The band at 34 kDa however disappeared indicating that this band was a 'real' **FP** target (that was now heat deactivated) while most other bands are more-or-less background labelling. Accordingly, an increase in temperature leads to more background labelling in *E. coli*.

As mentioned by the referee, the corresponding experiment for lysate labelling of *S. acidocaldarius* is reported in the Supplementary information (Supplementary Fig. 2). For these lysates, only a very low increase in background labelling can be observed (if at all). Accordingly and as expected from their adaption to high temperatures, *S. acidocaldarius* lysates are much less prone to unspecific labelling at high temperatures.

Although we do not aim to add the experimental data on *E. coli* labelling because we think that these findings do not really fit to the scope of the study, we nevertheless think that it is important to emphasize that the temperature-induced increase in background labelling for *S. acidocaldarius* is indeed only very moderate. We therefore changed the corresponding sentences in the manuscript to:

“Overall, we found comparable labelling patterns at 25 °C and 78 °C, even though some bands were slightly stronger and additional, very faintly labelled bands appeared at higher temperature indicating that higher temperatures induce an only weak increase in background labelling.”

3) One experiment that, if possible, I thought would be a nice way to determine the efficiency of the ABPP is to perform the esterase activity experiments described on page 15 and 16 in presence of the ABP to determine what percentage of overall esterase activity is blocked by the ABPP.

As suggested, we performed the respective experiments using the **NP** and **FP** (2 µM each) ABPs. The results are shown in Supplementary Figure 7.

The following sentences were added to the results section:

“Also, pre-incubation with ABPs **NP** and **FP** (2 μ M each) resulted in a nearly complete loss of esterase activity in the $\Delta\Delta$ mutant expressing the Saci_1105 (**Supplementary Fig. 7**). The **NP** concentration dependent exponential activity decay was determined and revealed an IC_{50} value of 0.15 μ M indicating high labelling efficiency of the ABP.”

These are my minor suggestions that I think would make this an even nicer manuscript. Textually, I think the manuscript would benefit from the removal of some rather long sentences, such as Page 4, lines 49-53, 54-56, page 5 86-89.

We have shortened (pg. 5, line 86-89) or split (pg. 4, lines 49-53 and 54-46) the criticized sentences.

In conclusion, I think this is an exciting paper that is very suitable for Nature Communications, as it presents a new application of in vivo ABPP to a very difficult class of enzymes.

Thank you for the compliment.

Reviewer #2 (Remarks to the Author):

The manuscript, Activity based protein profiling as a robust method for enzyme identification and screening in extremophilic Archaea, reports the development/demonstration of a robust and sensitive method to directly identify active serine hydrolases in extremophile cells in vivo. The method successfully identified serine hydrolases from thermoacidophilic archaea (*Sulfolobus*) and a halophile (*Haloferax*). The authors demonstrate that by combining the ABPP system with the genetic manipulation tools developed for *Sulfolobus acidocaldarius*, it is possible to directly screen for thermostable serine hydrolases that may be encoded in metagenome libraries. This was shown with two genes from a bacterial metagenome library and a serine hydrolase gene from a thermoneutrophilic bacterium (*Thermotoga*). The versatility of the method is clearly shown and should attract the broad attention of those engaged in microbiology, metabolism, enzyme screening, biotechnology and chemical biology.

Furthermore, the study should encourage the development of similar systems using host cells that exhibit different extremophilic properties. Genetic systems are available for a variety of halophiles (*Halobacterium*, *Haloferax*), thermoneutrophilic archaea (*Pyrococcus*, *Thermococcus*), Psychrophiles (*Shewanella*), Alkaliphiles (*Alkaliphilic Bacillus*) and piezophiles (*Thermococcus barophilus*), which have the potential to be utilized as host cells for in vivo screening.

The applicability and versatility of the method is clearly proven with numerous experiments. Experiments were carefully designed and technically sound. The manuscript is well written and was easy to read. Specific comments are as follows. Comments 1, 10, 11, 15, 18, 19, 20 are relatively more important than the others.

We thank the reviewer on this positive feedback on our study.

1. Abstract: the first two sentences can be combined and shortened so that the abstract sheds more light on the results of the manuscript.

Done accordingly.

2. Abstract last line: Is the term archaeal necessary here?

No, not really. We therefore removed it.

3. Line 119 constantly being developed?

The reviewer is right – this does not sound well (we wanted to emphasize that many research groups continue to develop novel probes). We removed the corresponding subclause.

4. Figure 1: define RG

RG stands for 'reactive group'. We have added the corresponding definition to the Figure legend.

5. Line 94: Spell out Sulfolobus

Done accordingly.

6. Lines 125-127: no need to parenthesize

Done accordingly.

7. Line 153: grows optimally?

Yes. We thus added the word 'optimally'.

8. The title of the manuscript shown in the Supplementary Information section differs to that of the main text. Please unify.

Done accordingly.

9. There are only 6 supplementary figures, not 8. Revise or remove "Content".

Done accordingly.

10. Supplementary Figure 2: Is the low molecular weight band in the Hp-NP lane relevant? It seems that it is not labeled at high temperature.

No, we do not think that this is biologically relevant.

The unexpected formation of 'bands' in the very low molecular range is a common observation in ABPP experiments. Their 'appearance' is thereby 'irreproducible' and difficult to predict. They can for example be formed by a click reaction between excess reagents (ABP + Rh-N3) or by uncontrolled cleavage during lysate preparation. Correspondingly, gel images are often cut in the lower molecular range (which we have not done) to focus the attention to the part of the gel that contains the 'relevant' labelling.

11. Line 193: Can the authors describe the term broadest in more detail? Does this mean the probe that labels the highest number of proteins, or are the authors subtracting Paraoxon plus from Paraoxon minus? Also please rephrase gold standard.

We meant that FP labels the highest number of proteins. We accordingly changed the expression to

“...NP \equiv , as well as with FP \equiv as the probe labelling most proteins.”

We removed the corresponding 'gold standard' phrase from the manuscript.

12. It is very difficult to read the domain labels on the right panel of Supplementary Figure 3. Please enlarge (the letters) or maybe just indicate the gene annotations.

Done accordingly. On this instance, we also enlarged the corresponding letters in the Fig. 3 and 4 of the main manuscript.

13. Supplementary Figure 4, perhaps the molecular size markers should be indicated separately for the left and right gels?

Done accordingly.

14. Line 262: period

Added accordingly.

15. The reviewer had difficulty understanding which data sheet corresponded to the Supplementary Tables.

We are not quite sure if we understand what the reviewer is driving at as we reckon that the confusion might be caused by the conversion of the excel sheet into a pdf format upon submission. The corresponding files however need to be viewed as original excel files.

The corresponding Supplementary Tables report the results of the MS experiments in the 'standard' format for this type of data. The corresponding data presentation has not been invented by us but it is the general format from researchers from the MS field for reporting their experimental data.

Accordingly, on Page 3 of the Supplementary Information file, we give an overview of the supplementary files.

Supplementary Table 1 contains the list of targets and proteins identified by LC- MS/MS for Figures 2d; 3a; 3c; 4a; 4c; and S3. For each Figure we added a new sheet which can be called by left-clicking the respective tab at the bottom of the Excel window.

Supplementary Table 2 contains the PFAM annotation for the three Archaea strains used. Supplementary Table 3 contains all information about primers and plasmids used.

16. Line 275: Please write *S. acidocaldarius* before 1105.

Done accordingly.

17. Is it correct that SSO2517, or EST, was labeled?

Yes, this is correct. EST is the descriptive protein/gene name while SSO2517 is an alternative gene name from the sequencing project. We decided to provide also these alternative gene names where available for consistency reasons. Some genes do not possess descriptive gene names (e.g. Amidase SSO2122).

18. Overall, the results seem to suggest that labeling intensity correlates well with total activity. Does this also apply for labeling efficiency? The reviewer could not really conclude on this. Is a more active protein labeled more, are the differences in labeling intensity due to the presence of more protein? If possible, this should be discussed.

We are not sure if we fully understand this question. Yes, labeling intensities correlate well with total activity and it is thus possible to compare the intensities of the same protein band between two samples to determine which sample contains more active enzymes (which is an approach commonly referred to as 'comparative ABPP').

However, ABPP measures activity *via* a binding event at the active site of an enzyme. Accordingly, not all active enzymes are labelled with the same intensity because even though they are active, the probe needs to bind to them first. The efficiency of binding however depends on the corresponding structural match between the probe and the target enzyme. This means that labelling intensity of bands from different enzymes with the same probe or from the same enzyme but with different probes does not necessarily need to fully correlate.

This overall means that each probe has a certain specificity towards its targets. A more specific probe will be less efficient in labelling 'some' targets and only really efficient in labelling its 'true' target(s). For example, **FP \equiv** is less specific and more efficient in labelling than **FP \equiv** . **FP \equiv** will label less proteins because it is more specific. It may label some targets of **FP \equiv** too but often the labelling will be weaker. Although we have not really investigated this, we think that this effect

might be caused by the different sterical demand of the two warheads: **FP** is sterically much less demanding than **NP**, it can thus bind to more active sites, whereas **NP** is bulkier and has thus more problems to reach certain active sites.

We however do not feel that a corresponding discussion should be added to our paper as these 'general considerations on ABPP' have been well and much more extensively discussed in general reviews on the ABPP approach. Some instructive reviews have been cited in the study.

19. Is the incubation time with NP= or FP= (2 h) an optimized time period? Is the effect of incubation time very large?

Indeed, we originally performed time course experiments with **FP** and **NP** on lysates but not on living cultures to evaluate a suitable incubation time. These experiments revealed that labelling of the major **FP** and **NP** target enzymes (1105 and 1116) was basically finished after 5 min while longer incubation times such as 2 h revealed the presence of additional bands. We therefore simply kept the 2 h incubation time also for the in vivo experiments.

20. Does the absence of 1105 and 1116 result in the enhanced labeling of any other proteins? Or is the concentration of probe already saturated under the applied conditions?

No, our gel-based assays do not give any evidence that a knock-out of 1105 or 1116 leads to increased labelling of the other or additional serine hydrolases.

We have not really investigated to which degree the probe gets consumed. However, we believe that there should be enough probe left to label any additional active serine hydrolase. This is based on a simple calculation. If we assume that the average molecular weight of the complex protein mix is 60 kDa and we perform labelling at a 1 mg/mL protein concentration, the corresponding average protein concentration is 16 μ M. We used the probes at a concentration of 4 μ M. Since serine hydrolases will be only a small part of the proteome, we are quite confident that the probe concentration is no limiting factor for labelling.

Reviewer #3 (Remarks to the Author):

The paper entitled "ABBP-based detection of native and heterologously expressed extremophile enzymes in Archaea" tries to use the "activity based protein profiling" to identify enzymes, more specifically serine hydrolases, and introduce it as a screening method for enzymes from extremophile microorganisms, in particular Archaea.

The authors have applied this method both in vitro and in vivo to identify serine hydrolases. In vitro they have used recombinant proteins and the results have been successful with esterases.

This method or technique can provide researchers working with archaea with a chemical tool to set out the assignment of functions to many hypothetical enzymes that find in archaeal genomes.

The work is well done and perfectly detailed to reproduce by others researchers.

The approach is very interesting but the most important concern about the applicability is the developing of novel ABPs to identify more enzyme classes.

The paper deserves to be published.

Thank you very much for this positive feedback.

REVIEWERS' COMMENTS:

Reviewer #1 (Remarks to the Author):

Thank you for your thorough address of the points raised. I am in full agreement with the assessment of the authors of the new data and cannot see any reason not to publish this excellent manuscript.

Reviewer #2 (Remarks to the Author):

The revised version of the manuscript, Activity based protein profiling as a robust method for enzyme identification and screening in extremophilic Archaea, has been sufficiently improved, and has fully addressed all concerns of this reviewer. The manuscript is sure to have a great impact in a broad range of research fields.

Point-by-Point response to reviewers' concerns

The reviewers did not raise any additional issues. Please see below their corresponding statements on the revision.

REVIEWERS' COMMENTS:

Reviewer #1 (Remarks to the Author):

Thank you for your thorough address of the points raised. I am in full agreement with the assessment of the authors of the new data and cannot see any reason not to publish this excellent manuscript.

Reviewer #2 (Remarks to the Author):

The revised version of the manuscript, Activity based protein profiling as a robust method for enzyme identification and screening in extremophilic Archaea, has been sufficiently improved, and has fully addressed all concerns of this reviewer. The manuscript is sure to have a great impact in a broad range of research fields.

We thank the reviewers for their reviewing efforts and positive feedback on our manuscript revision.